# CD38 and Anti-CD38 Monoclonal Antibodies in AL Amyloidosis: Targeting Plasma Cells and beyond

**DOI:** 10.3390/ijms21114129

**Published:** 2020-06-10

**Authors:** Dario Roccatello, Roberta Fenoglio, Savino Sciascia, Carla Naretto, Daniela Rossi, Michela Ferro, Antonella Barreca, Fabio Malavasi, Simone Baldovino

**Affiliations:** 1Nephrology and Dialysis Unit & CMID (Center of Research of Immunopathology and Rare Diseases), Coordinating Center of the Network for Rare Diseases of Piedmont and Aosta Valley, San Giovanni Bosco Hub Hospital of Turin, and Department of Clinical and Biological Sciences, University of Turin, 10154 Turin, Italy; roberta.fenoglio@unito.it (R.F.); savino.sciascia@unito.it (S.S.); carla.naretto@alscittaditorino.it (C.N.); daniela.rossi@unito.it (D.R.); michela.ferro@alscittaditorino.it (M.F.); simone.baldovino@unito.it (S.B.); 2Pathology Division, Department of Oncology, University of Turin, 10154 Turin, Italy; antonella.barreca@unito.it; 3Department of Medical Science, University of Turin, and Fondazione Ricerca Molinette, 10154 Turin, Italy; fabio.malavasi@unito.it

**Keywords:** AL amyloidosis, CD38, anti-CD38 MoAb, Daratumumab, Isatuximab

## Abstract

Immunoglobulin light chain amyloidosis (AL amyloidosis) is a rare systemic disease characterized by monoclonal light chains (LCs) depositing in tissue as insoluble fibrils, causing irreversible tissue damage. The mechanisms involved in aggregation and deposition of LCs are not fully understood, but CD138/38 plasma cells (PCs) are undoubtedly involved in monoclonal LC production.CD38 is a pleiotropic molecule detectable on the surface of PCs and maintained during the neoplastic transformation in multiple myeloma (MM). CD38 is expressed on T, B and NK cell populations as well, though at a lower cell surface density. CD38 is an ideal target in the management of PC dyscrasia, including AL amyloidosis, and indeed anti-CD38 monoclonal antibodies (MoAbs) have promising therapeutic potential. Anti-CD38 MoAbs act both as PC-depleting agents and as modulators of the balance of the immune cells. These aspects, together with their interaction with Fc receptors (FcRs) and neonatal FcRs, are specifically addressed in this paper. Moreover, the initiallyavailable experiences with the anti-CD38 MoAb DARA in AL amyloidosis are reviewed.

## 1. Introduction

Systemic amyloidosis is characterized by abnormal production and deposition in the extracellular space of misfolded proteins, resulting in a heterogeneous spectrum of clinical conditions [1]. The most prevalent type, namely immunoglobulin light chain amyloidosis (AL amyloidosis), is associated with deposition in the targeted organs of the light chains (LCs) of the immunoglobulins [2]. AL amyloidosis is a rare disease with an incidence of about 1 person/million/year. Due to its rarity and non-specific presentation, diagnosis is often late and frequently occurs after one year from initial symptom presentations. AL amyloidosis can be detected in 30% of patients newly diagnosed as having MM, but it mostly complicates monoclonal gammopathies of undetermined significance [1], which have a 10-foldlower relative risk of developing AL amyloidosis. The clinical manifestations of AL amyloidosis depend on organ involvement. However, the diagnosis can be challenging as symptoms might mimic other more frequent conditions. The deposition of monoclonal light-chain proteins in AL amyloidosis can induce toxic damages in several organs, with the heart and kidney being most frequently affected [3]. 

AL amyloidosis is often associated with a poor prognosis, with patients having a mean survival ranging from six months to three years, according to the characteristics of the investigated cohort [4,5]. The degree of cardiac involvement represents a major determinant of the outcome in patients with AL amyloidosis, with up to a third of patients with severe cardiac damage having a fatal outcome within 12 months from diagnosis [1,6].

Renal involvement, as identified by the detection of decreased estimated glomerular filtration rate (eGFR) or the presence of proteinuria, is found in approximately 70% of patients [7,8,9,10,11,12,13]. The risk of dialysis at two years is 11%–25% in patients with eitherdecreased eGFR orproteinuria, and up to 60%–75% in patients with both decreased eGFR and proteinuria [14,15].

The goals of therapy should be to suppress the production of the pathologic LC precursor and lessen organ impairment. The latter is hard to achieve since the process of amyloid deposition is often irreversible [16]. Therefore, an effective treatment should be applied as soon as possible, before irreversible damage occurs. Only half of the patients treated with conventional regimens show normalization of LCs levels in serum (i.e., complete hematologic response) [17]. The absence of a complete hematologic response results in further deposition of amyloid and reduces the chances of the improvement of affected organs. Therefore, the standard escalation treatment in the attempt to control the hematological disorder (which is a milestone of conventional treatment) should not be applied to patients with rapid disease progression, such as those with renal involvement. Indeed, it could result in a delay in the effective management of the disease and in the consequent accumulation of irreversible lesions. These patients should be treated aggressively ab initio, and the availability of an effective target therapy is desirable. 

The involvement of the CD138+ 38+ monoclonal PCs in LCs production is well established, and CD38 could be considered a suitable target of PCs. 

The emerging therapeutic potential of anti-CD38 MoAbs in PC dyscrasias is addressed in this paper, and the initiallyavailable experiences with anti-CD38 MoAbs in AL amyloidosis are reviewed.

## 2. Evidence Supporting CD38 as an Ideal Target for Treating AL Amyloidosisand thePossible Therapeutic Role of Anti-CD38 Antibodies

Most of the available information about the therapeutic effects of targeting CD38 derives from studies on MM. AL amyloidosis and MM are both PC dyscrasias and share some genetic aberrations and therapeutic approaches; however, AL amyloidosis has a distinct phenotype and different prognostic features. The experience with anti-CD38 compounds in MM is critical for the development of novel strategies of management of AL amyloidosis. Indeed, the high expression of CD38 on the clonal PC surface represents the basis for target therapy.

Multiple roles have been described for CD38, as a receptor, adhesion molecule and ectoenzyme [18]. Due to these characteristics, CD38 could represent an effective molecule to be targeted by therapeutic antibodies in the management of AL amyloidosis, a pathologic condition characterized by high expression of this molecule at the cell surface level [19]. CD38 is not expressed by early stem cell progenitors [20]. Instead, B cells, activated T cells and NK express CD38 on their surface, making those cells a potential target for the anti-CD38 MoAbs [21,22,23,24,25,26,27,28,29,30,31]. 

Daratumumab (DARA), a fully human monoclonal antibody targeting CD38, is the first therapeutic anti-CD38 moAb clinically approved by the Food and Drug Administration for the management of relapsing MM. Its use has been approved in monotherapy as well as in association with lenalidomide or bortezomib. After DARA was approved, more anti-CD38 MoAbs have been reported, such as isatuximab (ISA) and MOR202 [32].

In order to speculate on the potential effect of anti-CD38 antibodies in AL amyloidosis, the mechanisms through which DARA (the most analyzed anti-CD38 agent) exerts its cytotoxic role on effector cells in MM (that is mainly mediated by anFcR-dependent mechanism) should be mentioned.

The principal effects of DARA include antibody-dependent cellular cytotoxicity (ADCC), antibody-dependent cellular phagocytosis (ADCP), complement-dependent cytotoxicity (CDC), and direct apoptosis after secondary cross-linking [20].

ADCC is induced by the release of cytotoxic cytokines and cellular mediators (to include perforins and granzymes) by NK cells. Immunomodulatory imide drugs (IMiDs) have a synergic effect when combined with anti-CD38 moAb by implementing the activity and the number of NK cells [31,32]. However, after treatment with anti-CD38 moAb, a paradoxical reduction of NK cells can be found, possibly due to cytotoxic crosstalk between NK cells and DARA [33]. Whether this mechanism reduces the synergic effect of ADCC in the context of MM is still under debate [34]. 

ADCP is a further potent mechanism through which DARA can exert its effect by inducing a trigger effect on monocytes and macrophages targeting antibody-opsonized MM cells. It has been speculated that the proportion of monocytes and MM cells can play a role in DARA-mediated antibody-dependent cellular phagocytosis. In this context, the mechanisms supporting the contribution of the CD47 pathway in the modulation of phagocytosis by monocytes has been investigated, showing that an anti-CD47 antibody might potentiate the ADCP effect induced by DARA [35]. 

CDC follows the ligation of DARA on CD38 on MM cells [29]. DARA is reportedly the most efficient among anti-CD38 MoAbs in triggering the classical complement pathway [29,36,37,38].

Moreover, CD38 has an ectoenzyme activity, associated with the release of extracellular adenosine (ADO) [39]. Several biological functions have been described for ADO, to include the ability of the nucleoside to exert an inhibitory effect on the immune system through modulation of the activity of several cells populations (e.g., NK cells, monocytes, dendritic cells, T and B lymphocytes, and macrophages) [39]. The potential additional therapeutic effect of targeting ADO in order to obtain an inhibition of the immune system is still under investigation. 

Furthermore, DARA is able to reduce the expression of CD38 on the cell surface by trogocytosis. This effect is based on a switch of the CD38-antiCD38 MoAb complex between the abnormal PCs, monocytes, and neutrophils. Again, the process results in an inhibition of ADO levels, with a consequent effect on the tolerogenic microenvironment [40]. 

DARA also induces polarization and redistribution of CD38 on the myeloma cell membrane surface, resulting in the release of microvesicles expressing the CD38-antiCD38 complex. The role of these changes is not elucidated yet [20]. 

DARA also exerts a direct immunomodulatory activity on immune cells [13,14,15,16,17,18,19,20,21,22,23,24,25,26,27,28,29,30,31,32,33,34,35,36,37,38,39,40,41] and is able to deplete CD38+ suppressor cells, namely B*reg*, T*reg* and myeloid-derived suppressor cells. Remarkably, along with the known effect on T*reg* and myeloid-derived suppressor cells, emerging evidence is supporting the theory that the MM microenvironment supports the survival of B*reg*, leading to an overall immunosuppression effect [42]. 

Another recently launched anti-CD38 MoAb is ISA. This MoAb recognized a specific epitope on human CD38. Its molecular target is represented by a completely different amino acid sequence when compared to DARA [36,37]. ISA has been shown to exert both a potent pro-apoptotic effect, regardless of the presence of cross-linking agents, and robust ADCC (the most prevalent effector mechanism for the elimination of tumor plasma), CDC and ADCP against CD38+ malignant subpopulations [8]. 

Interestingly, a direct association between the level of CD38+ expression and mechanisms activated by ISA has been found in preclinical models [43,44].

ISA has also been shown to exert its immunomodulatory activity by reducing CD38+ T*reg* and, at the same time, by potentiating NK cells and T lymphocytes-mediated immune response [43,45,46]. Moreover, ISA has inhibitory effects on immune-checkpoint molecules, such as PD-L1 on osteoclasts [41,42,43,44,45]. 

MOR202 is a fully human anti-CD38 antibody that is currently under investigation in phase I/IIa clinical trials in MM. The ability of MOR202 of inducing both ADCC and ADCP effects in MM cells makes this molecule a promising candidate for new therapeutic regimens in the management of patients with MM [47]. As for the other anti-CD38 moAbs, the cytotoxic effect of MOR202 on MM cells is augmented by IMiD compounds, such as lenalidomide and pomalidomide. These compounds, apart from being involved in the activation of effector cells and direct cytotoxicity, are able to upregulate CD38. These mechanisms represent an indication for combining MOR202 with IMiD compounds.

Each of these three anti-CD38 MoAbs provided a strong case for being used in AL amyloidosis. However, the effects of ISA are strongly related to CD38 expression (which could be a drawback in the AL amyloidosis setting as this condition is characterized by a small burden of abnormal PCs) and both ISA and MOR202 seem to require IMiD co-operation to achieve an optimal effect. This might restrict their use in very co-morbid patients, such as subjects with AL amyloidosis. Moreover, insights of efficacy and safety in AL amyloidosis are presently limited to DARA. 

## 3. From Basic Research to Clinical Application in AL Amyloidosis: Available Experiences

As previously emphasized, the relatively small percentage of clonally restricted plasma cells in AL amyloidosis expresses CD38, suggesting anti-CD38 MoAbs to be putatively effective in this disease [28]. 

DARA is the only anti-CD38 MoAb that has been formally examined over the last few years for the treatment of AL amyloidosis [46,48,49,50,51,52]. Nevertheless, information about organ improvement, especially the kidney, suffers from imprecise criteria of the definition of organ involvement. 

Sanchorawala et al. showed high hematologic response rates (>80%) in 21 patients with relapsed AL amyloidosis [46]. No data were available on renal response.

Roussel et al. examined 84 AL amyloidosis patients who were given DARA either in combination with dexamethasone or other plasma-cell-directed therapies. Eighty-four percent of the patients had a hematologic response, in the majority of cases within one month. Several patients had cardiac involvement, and half of them showed a cardiac response within two months. Only 26 of the 53 patients with renal impairment or urinary abnormalities were evaluable. They showed some renal response within six months [50]. Unfortunately, none of the patients in this series were reported as having biopsy-proven renal involvement. As far as renal implications are concerned, the identification of amyloid deposits represents the only proof of kidney involvement. Moreover, the entity and distribution of renal amyloid deposition might also be important when comparing the outcome of these patients [51]. 

In a multicenter phase II study on DARA monotherapy [50], 40 patients from 15 centers, including 26 patients with presumptive renal involvement, who were previously treated with other agents, had been examined. This is another example of misinterpretation and confounding data, occurring when nephrologists are not involved in data evaluation. Indeed, no renal biopsies had been carried out and definitions of renal involvement were not provided. Twenty-one of these patients had <60 mL/min/ 1.73 m^2^ eGFR. This figure, considering patients’ mean age (69 years), is close to normal. Seven patients were defined as having had a renal response because of a 30% decrease in proteinuria without a 25% percent increase in eGFR. 

With regard to renal response, these results are difficult to interpret.

We attempted DARA monotherapy in four severe cases of AL with multiorgan and biopsy-proven renal involvement. Two males and two females (mean age 64 years, ranging from 52 to 69) were treated with DARA following antibody testing and extended RBC antigen phenotyping. The treatment protocol included 16mg/kg DARA administered intravenously weekly for eight consecutive weeks, then every two weeks for another eight administrations, and lastly, monthly until the 52nd week. One patient was refractory to conventional schedules, one was treated for relapsing disease, one was intolerant and one was treated front-line. Administration of DARA resulted in the disappearance of serum M-component and Bence–Jones proteinuria, and normalization or improvement of the free light chain ratio with a decrease in N-terminal pro-peptide levels and a dramatic drop in urinary protein loss. Cytofluorimetric profiles showed complete disappearance of peripheral PCs with a decrease in both NK and B cell CD19+ve and a slight increase in T helper cells.

## 4. Expanding the Role of CD38: Future Perspectives

CD38 is identifiable on several non-pathological cell subpopulations, including NK cells, B lymphocytes and activated T cells. Therefore, anti CD38 MoAbs could also potentially exert an effect on non-pathological cells [20]. On the other hand, several investigations showed that anti-CD38 MoAbs can trigger a depletion of CD38+ immunosuppressive cells, including T*reg*, B*reg* and myeloid-derived suppressor elements cells [1,42]. These observations further support the rationale of an anti-CD38 MoAb-based regimen as a therapeutic tool for PC dyscrasias. 

Further considerations are worth mentioning when considering expanding the potential indication for anti-CD38 MoAbs. 

As NK cells mediate ADCC, these cells have a main role in enhancing the activity of anti-CD38 MoAbs [53,54,55,56,57]. DARA increases NK-cell cytotoxicity against cells expressing high, but not low, CD38 [50]. This could be used as a platform for a new therapeutic target for CD38+ cells beyond PC. Besides, in a syngeneic in vivo tumor model neoplasia study, the therapeutic effect of DARA was shown to trigger programmed cell death of myeloma cells via a cross-linking mechanism [20,53,54,55,56,57]. Intriguingly, the crosstalk between DARA and FcRs seems to play a pivotal role in triggering the activity of the MoAbs (Figure 1). The decreased levels of NK cells found in subjects treated with DARA could be due to an antibody-mediated fratricide between NK cells. However, NK cells reduction does not significantly affect DARA efficacy [58]. Moreover, this effect can be balanced by agonistic agents. For instance, IMiDs can have a synergic effect with MoAbs directly targeting CD38 (not limited to DARA) by bursting the activity of NK cells, with a consequent increase in the ADCC [59,60]. Similarly, ADCP is a further mechanism through which anti-CD38 mAbs exert their action on monocytes and macrophages by antibody-opsonized cells [61]. The proportions of monocytes and abnormal PCs could impact on DARA-mediated ADCP [61]. Moreover, the CD47 pathway has been shown to regulate monocyte-driven phagocytosis, and anti-CD47 antibody has been found to increase the ADCP activity triggered by DARA [62]. 

With regard to other potential agents upcoming as alternative options of anti-CD38 target therapy, ISA exerts both robust pro-apoptotic activity and strong ADCC-associated anti-neoplastic effects, ADCP and CDC. Several additional mechanisms have been described, including (a) homotypic aggregation-associated cell death (as observed in MM cells) that is influenced by the expression of CD38 on the cell surface and is related to the actin cytoskeleton and membrane lipid rafts [39]; (b) up-modulation of reactive oxygen species; (c) lysosome-mediated cell death via alteration of the lysosomes structure and upregulation of the lysosomal membrane permeability; (d) caspase 3 and 7-mediated apoptosis induced in cells highly expressing CD38 (typically MM PCs). 

Research aimed at examining in depth the effects of the antibody binding to CD38 is crucial. It has been observed that the process of interaction between CD38 and anti-CD38 MoAbs can influence either internalization or externalization of the target/antibody complex, with a consequent effect on the release of microvesicles [53,54,55,56,57]. For instance, DARA creates a polarization and redistribution of CD38 on the MM cell surface and the development and shedding of microvesicles rich in CD38 bind to DARA in biological fluids. These microvesicles differ when compared to those spontaneously released. Indeed, they present antibody on their surface, express ectoenzymes able to metabolize adenosine triphosphate and nicotinamide adenine dinucleotide to produce adenosine. Besides, they have the potential to merge with neighboring cells and eventually escape the myeloma niche, reaching the blood. 

Albeit the process has still to be fully elucidated, microvesicles can also undergo a process of uptake into the cytoplasm of myeloid-derived suppressor cells, NK and monocytes. Pilot in vitro data have shown that microvesicles derived from MM cells exposed to DARA might be able to modulate gene expression at the level of the immune response in purified NK cells. 

The effects of microvesicles on dendritic cells for possible vaccinal effects are currently under evaluation.

Another point to be considered as relevant for anti-CD38 MoAbs efficacy is the activation of CDC [63,64]. Among anti-CD38 moAbs, DARA has been proved to be the stronger activator of the classical complement cascade, whilst MOR202 shows a moderate CDC activity [63]. Instead, the greater direct pro-apoptotic effect has been associated with ISA, regardless of the cross-linking, and seems to be exerted through the activation of caspases 3 and 7 [65]. The latest effect was not reported for DARA and MOR202 and it might depend on the different epitope recognition in the target CD38 molecule by the different MoAbs [32,66]. When referring to the putative immunomodulatory effects following direct targeting of immune cells, DARA is able to deplete CD38+ immune-suppressor cells such as T*reg* and myeloid-derived suppressor cells [67]. Incidentally, emerging evidence shows that the MM microenvironment supports the survival of B*reg* which, in turn, exerts an immunosuppressive effect [41]. ISA is also capable of immunomodulatory effects. It decreases CD38+ T*reg* and increases NK cell activity proportionately [43]. ISA has also been shown to have an inhibitory effect on numerous immune-checkpoint molecules.

How the majority of these effects can enhance the efficacy of anti-CD38 compounds in AL amyloidosis is currently only speculative.

## 5. Conclusions

The understanding of the pathogenesis of tissue damage in AL amyloidosis has remarkably improved in recent years. Nevertheless, this condition, especially when heart and kidneys are involved, continues to have an unacceptably poor prognosis. Amyloid deposition is often a permanent process for which putatively effective treatments should be used in a timely manner, before irreversible damage has been established. Autologous stem cell transplantation is thought to be the most definite PC-directed therapy in AL amyloidosis [1]. However, due to the delay in diagnosis and the extent of cardiac involvement, only a minority of patients are eligible for transplantation. Consonant with this observation are our data from fifty-two AL amyloidosis subjects followed at our Center between 2007 and 2018. As many as thirty-one were ineligible for bone marrow transplantation. Therefore, alternative therapies with a high degree of safety due to the burden of co-morbidities of these patients are urgently needed. A better understating of the mechanisms underlying the regulation of PC survival in PC dyscrasia and of the crosstalk with the immune microenvironment have promoted the description of new target therapies.

CD38 target therapy should both deplete and modulate immune cells. While the majority of in vitro and in vivo observations have been done in MM, initial experiences might be a launching pad for expanding research on the use of anti-CD38 target therapy in AL amyloidosis. Different anti-CD38 MoAbs have been designed to target abnormal PCs via the Fc-dependent immune effector mechanism. Among the most widely used anti-CD38 compounds in an MM setting DARA is a full human MoAb while ISA is chimeric. It is rational to suppose that the dissimilarities in structure between DARA and ISA explain the diverse interactions with the FcRs and the different molecular mechanisms involved in their interaction with the target molecule. Available data on the use of anti-CD38 MoAbs, especially in AL amyloidosis, are pivotal, but pave the way to studies aimed at characterizing novel therapeutic protocols.

AL amyloidosis is characterized to a limited extent by abnormal PCs producing a huge amount of light chains susceptible to aggregation in insoluble form and deposition in target organs. Causal therapy is mainly addressed to interrupt the synthesis of abnormal proteins. It should be remembered that the extent of PC dyscrasias widely differs between MM and AL amyloidosis. Combination therapies (i.e., anti-CD38 MoAbs plus IMiDs and/or proteasome inhibitors) are needed to lessen the tumor burden in MM, and the rate of relapses justifies an escalation approach to improve patient survival. As compared to MM, management of AL amyloidosis probably needs less intensive treatment, due to the limited dimension of the clone. However, treatment should be extremely timely, maybe in an upfront setting, in order to prevent definitive organ damage. Target therapy with anti-CD38 MoAbs in AL amyloidosis could be revealed to be the most appropriate strategy, even when given alone, to reduce the adverse effects in these fragile patients, maybe at a personalized dose, and perhaps for a more prolonged time. 

Finally, the experience with novel protocols for AL amyloidosis could be transferred to other diseases. Apart from incorporating the approaches used in MM into future strategies able to reduce the mortality of AL amyloidosis patients, the challenge for the near future will be to design novel therapeutic protocols for each disorder attributable to a PC dyscrasia. These conditions include those with a small clone of PCs producing harmful light chains causing irreversible organ damage distally to the production site. Ideally, a number of rare diseases could benefit from therapeutic schemes validated in AL amyloidosis i.e., monoclonal immunoglobulin deposition disease, proliferative glomerulonephritis with monoclonal IgG-K deposits, type I cryoglobulinemia with clonally restricted IgG, light chain nephropathy and fibrillary glomerulonephritis with a monotypic light chain.

## Figures and Tables

**Figure 1 ijms-21-04129-f001:**
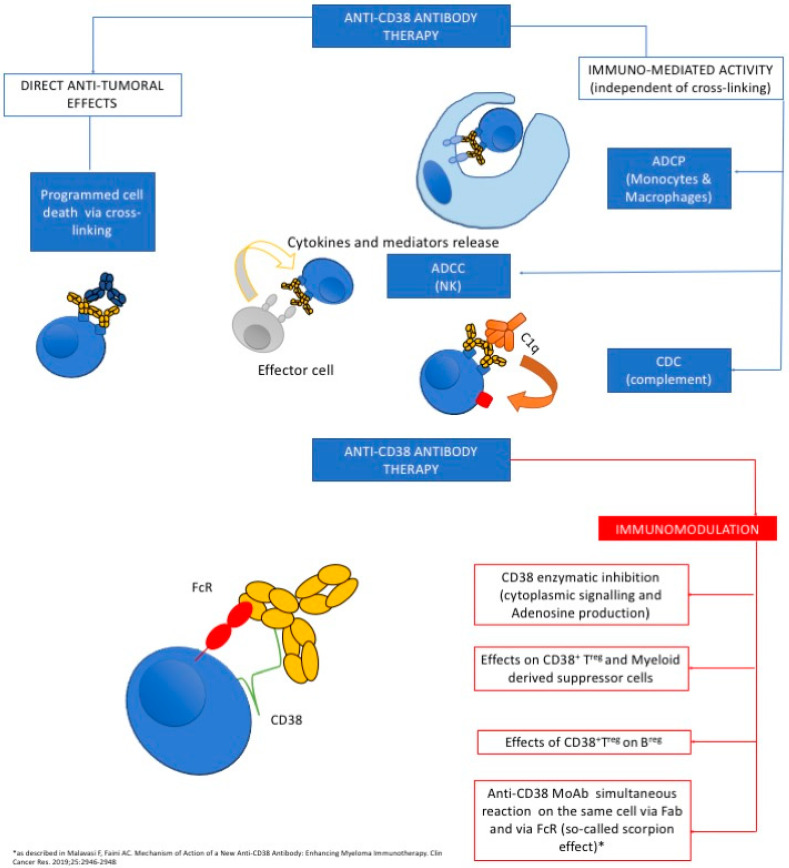
(UPPER PANEL) The left side of the figure illustrates the key effects of the anti-CD38 antibodies on the tumor target and on the main functional effector cells. On the right, a diagram of a hypothetical extension of the immunomodulatory effects mediated by anti-CD38 antibodies. The most intriguing hypothesis is based on the functional synergic interactions between the antibodies and their IgG Fc receptors expressed at various levels by myeloid and lymphoid effectors. The diagram (LOWER PANEL) also shows that the anti-CD38 antibodies may react simultaneously on the same cell via Fab and via FcR, through the so-called scorpion effect [54].

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
