# Peer review of "CD38 and Anti-CD38 Monoclonal Antibodies in AL Amyloidosis: Targeting Plasma Cells and beyond"

_ijms, 2020, doi:10.3390/ijms21114129_

Round 1

Reviewer 1 Report

The work makes a valuable contribution to timeous treatment of AL amyloidosis. It is comprehensive and well referenced and I would recommend publication. However, the English makes it difficult to read in certain instances for example, in the Introduction - where "specialists visit the patients"?

I think that where "in vivo"  is mentioned, it should refer to clinical studies as in vivo usually refers to animal studies. 

Table 1 is confusing as the information is duplicated. 

Author Response

Reviewer #1: 

R1.1: The work makes a valuable contribution to timeous treatment of AL amyloidosis. It is comprehensive and well referenced and I would recommend publication. However, the English makes it difficult to read in certain instances for example, in the Introduction - where "specialists visit the patients"?

Reply1.1. Thank you for your generally favorable comments. The manuscript has been edited by a native English speaker to improve the readability. 

R1.2 I think that where "in vivo"  is mentioned, it should refer to clinical studies as in vivo usually refers to animal studies. 

Reply 1.2: the manuscript has been checked an amended as suggested when necessary.

R1.3. Table 1 is confusing as the information is duplicated. 

Reply 1.3 To reduce redundancy, Table 1 has been omitted.

Reviewer 2 Report

This is a well written review. Covered all relevant and important aspects. Concluding remarks were appropriate.

Author Response

This is a well written review. Covered all relevant and important aspects. Concluding remarks were appropriate.

Reply2.1. Thank you for your favorable comments.

Reviewer 3 Report

The authors describe the comprehensive review of CD38 associated with AL amyloidosis. A couple of comments can be made to more improve the manuscript for readers.

Minor

  • Results are explained well in accordance with lots of papers. However, each item feels quite fragmentary.
  • There are a lot of misspelling and incomplete reference.

Author Response

Reviewer #3

Results are explained well in accordance with lots of papers. However, each item feels quite fragmentary. There are a lot of misspelling and incomplete reference.

Reply3.1. Thank you for your generally favorable comments. The manuscript has been edited by a native English speaker to improve the readability. References' list has been amended when necessary.